# Evidence of Mitochondrial Dysfunction in Fibromyalgia: Deviating Muscle Energy Metabolism Detected Using Microdialysis and Magnetic Resonance

**DOI:** 10.3390/jcm9113527

**Published:** 2020-10-31

**Authors:** Björn Gerdle, Bijar Ghafouri, Eva Lund, Ann Bengtsson, Peter Lundberg, Helene van Ettinger-Veenstra, Olof Dahlqvist Leinhard, Mikael Fredrik Forsgren

**Affiliations:** 1Pain and Rehabilitation Centre, and Department of Health, Medicine and Caring Sciences, Linköping University, SE 581 83 Linköping, Sweden; bijar.ghafouri@liu.se (B.G.); annbengtsson@yahoo.se (A.B.); helene.veenstra@liu.se (H.v.E.-V.); 2Center for Medical Image Science and Visualization (CMIV), Linköping University, SE 581 83 Linköping, Sweden; peter.lundberg@liu.se (P.L.); olof.dahlqvist.leinhard@liu.se (O.D.L.); mikael.f.forsgren@liu.se (M.F.F.); 3Radiation Physics, Department of Health, Medicine and Caring Sciences, Linköping University, SE 581 83 Linköping, Sweden; eva.lund@liu.se; 4Radiation Physics, Department of Health, Medicine and Caring Sciences, SE 581 83 Linköping University & Department of Radiation Physics, UHL, SE 581 85 Linköping, Sweden; 5AMRA Medical AB, SE 582 22 Linköping, Sweden

**Keywords:** fibromyalgia, chronic pain, muscle, microdialysis, ATP, PCr, magnetic resonance spectroscopy, magnetic resonance imaging

## Abstract

In fibromyalgia (FM) muscle metabolism, studies are sparse and conflicting associations have been found between muscle metabolism and pain aspects. This study compared alterations in metabolic substances and blood flow in erector spinae and trapezius of FM patients and healthy controls. FM patients (*n* = 33) and healthy controls (*n* = 31) underwent a clinical examination that included pressure pain thresholds and physical tests, completion of a health questionnaire, participation in microdialysis investigations of the etrapezius and erector spinae muscles, and also underwent phosphorus-31 magnetic resonance spectroscopy of the erector spinae muscle. At the baseline, FM had significantly higher levels of pyruvate in both muscles. Significantly lower concentrations of phosphocreatine (PCr) and nucleotide triphosphate (mainly adenosine triphosphate) in erector spinae were found in FM. Blood flow in erector spinae was significantly lower in FM. Significant associations between metabolic variables and pain aspects (pain intensity and pressure pain threshold PPT) were found in FM. Our results suggest that FM has mitochondrial dysfunction, although it is unclear whether inactivity, obesity, aging, and pain are causes of, the results of, or coincidental to the mitochondrial dysfunction. The significant regressions of pain intensity and PPT in FM agree with other studies reporting associations between peripheral biological factors and pain aspects.

## 1. Introduction

Pain arises in the brain because of influences from neurobiological, psychological, and social/contextual aspects and their complex interactions. Fibromyalgia (FM) is characterized by chronic widespread pain and often generalized hyperalgesia/allodynia. Depending on the diagnostic criteria, FM has a prevalence of 2–8%. FM is associated with high comorbidity for other chronic pain conditions and somatic and psychological symptoms/disorders. No definite pathophysiology has been established. Imaging techniques have challenged previous ideas about the peripheral origin of FM and have provided evidence for altered central nervous system (CNS) nociceptive/pain processing and morphology in FM. However, recent studies have reported both central alterations and peripheral alterations (e.g., systemic low-grade inflammation and nociceptor/muscle alterations) [1,2,3,4,5]. Understanding these complicated peripheral and central processes and their interactions are fundamental for developing treatments.

In real life muscle work, several fuels are used to produce adenosine triphosphates (ATPs), the cell’s energy currency [6]. The fuels—mainly carbohydrates (plasma glucose and muscle glycogen) and fats (plasma free fatty acids and muscle triglycerides), but also amino acids—are used by muscle cells to produce ATPs [6,7,8].

To analyze muscle metabolism, several FM studies have used microdialysis (MD), an in vivo technique, to collect substances from the muscle interstitium. Two MD studies of FM reported increased concentrations of lactate and/or pyruvate (i.e., products of glycolysis) in the trapezius [9,10]. Another FM study found that the concentrations of both lactate and pyruvate were significantly increased in the vastus lateralis [11]. One study of the trapezius found significant associations between these substances and pain intensity and PPT [10]; however, the two other studies failed to establish such significant correlations [9,11].

^31^P magnetic resonance spectroscopy (^31^P-MRS) can be used to non-invasively investigate phosphorous metabolism, carbohydrate metabolism, and intracellular pH in muscles [12] (i.e., mitochondrial function [13]). This method has been used by several FM studies, but findings are inconsistent (see Gerdle et al. [14]). This inconsistency is probably due to differences in the conditions investigated, differences in the type of muscle investigated, and whether absolute or relative measures of metabolism were used. The only two studies that determined absolute muscle concentrations of adenosine triphosphate (ATP) and phosphocreatine (also known as creatine phosphate (CP)) (PCr) in FM reported significantly lower levels in the quadriceps [14,15]. One of these studies found an association between ATP concentration and pain intensity just above the significance threshold (*p* = 0.057) [15], whereas the other study found no correlation between ATP and PCR and pain intensity [14].

This study is motivated by the fact that muscle metabolism is poorly investigated in FM patients, many FM patients have reported that their pain condition started and persists in neck-shoulder muscles and low back muscles, and that findings from previous studies are inconsistent with respect to FM associations with central factors such as pain intensity and PPT. To address these shortcomings, we compared alterations in metabolic status (including blood flow) in the erector spinae and trapezius of female FM patients and matched healthy controls. In addition, we investigated to what extent muscle metabolic status correlated with PPT, pain intensity, and physical performance. Thus, this relatively large study using different methods to capture the muscle metabolic status has potential to shed further light on pathophysiology in FM (i.e., possible mitochondrial involvement). Moreover, this study applied advanced multivariate methods, which can determine is muscle metabolic alterations are associated with pain aspects in FM with better precision than earlier studies.

## 2. Subjects and Methods

### 2.1. Subjects

This large-scale investigation of FM included 33 female FM patients and 31 age-matched female controls (CON) between 22 and 56 years old. The CON group was recruited through advertisements in newspapers, and the FM group was recruited from former patients at the Pain and Rehabilitation Center at the University Hospital in Linköping. The number of subjects was determined using Power and Sample Size Calculation ver 3.0.2 [16] based on results from both microdialysis (the concentration of lactate according to Rosendal et al. [17]) and from ATP concentration obtained from spectroscopy of the vastus lateralis (Gerdle et al. [14]). Both analyses indicated that 25 subjects in each group were necessary. Of the 64 subjects recruited to this study, 62 participated in the ^31^P-MRS part and 57 participated in the MD part of the study. Details on the inclusion and exclusion criteria as well as clinical examination are given below.

The study was granted ethical clearances by Linköping University Ethics Committee (Dnr: 2016/239-31). All participants gave their written informed consent, and the study was performed in accordance with the Helsinki Declaration.

### 2.2. Procedures

At the first visit, the subjects underwent a clinical examination that included pain threshold tests and physical tests. In addition, the subjects completed a health questionnaire that covered aspects of pain, health, disability, demographic data, and psychological characteristics. At the second visit, the subjects underwent MD of the trapezius and the erector spinae muscles. At the third visit, spectroscopy was used to evaluate the subjects’ erector spinae.

### 2.3. Clinical Examinations

Both FM patients and controls underwent a brief clinical examination of the heart and lungs, which included recording diastolic and systolic blood pressure (mm Hg) after two minutes of rest. In addition, their weight (kg) and height (m) were recorded. Using these measurements, Body Mass Index (BMI, kg/m^2^) was calculated and classified according to the World Health Organization (WHO) criteria: <18.5 = underweight; 18.5–24.9 = normal range; 25.0–29.9 = overweight; 30.0–34.9 = obesity; and ≥35.0 = severe obesity. The clinical examination ensured that the controls were healthy with respect to anamnesis for rheumatic diseases, neurological diseases, diabetes, cardio-vascular diseases, psychiatric diseases, and high alcohol consumption (i.e., Alcohol Use Disorders Identification Test (AUDIT) >6 according to the recommendations for women).

The clinical examination of the patients ensured that they met the criteria for FM according to the 1990 criteria from the American College of Rheumatology (ACR) [18] and that they were not abusing alcohol (i.e., high alcohol consumption) according to AUDIT. The number of tender points were registered both in FM and in CON. The ACR criteria from 1990 is based on anamnestic reports and semi-objective examinations of hyperalgesia/allodynia (tender points). Newer criteria (2010/2011 and 2016) based on anamnestic reports have been established [19,20,21]. As we wanted to compare our study with earlier studies, we chose the ACR criteria from 1990.

#### 2.3.1. Pressure Pain Thresholds

Pressure pain thresholds (PPT) were determined using a manual pressure algometer (Somedic AB, Sweden) mounted with a probe (contact area of 1 cm^2^) on the muscle belly (for details, see [22,23]). The erector spinae, tibialis anterior, and trapezius were investigated bilaterally. The pressure was increased by 30 kPa/s until the subject perceived pain, indicated by pushing a stop button or until the maximum threshold of 600 kPa was reached. The PPT for each anatomical location was defined as the mean of two trials obtained with a minimum interval of 30 s. We used the mean of the six anatomical locations (PPT-tot) as well as PPT for the trapezius and erector spinae (i.e., the muscles that were assayed using MD).

#### 2.3.2. Physical tests

##### Hand function

Grip force (N) was measured using Grippit (AB Detektor, Gothenburg, Sweden). Peak value (Grip force-max), average value (Grip force-average), and 10-s value (Grip force-endur) were recorded for 10 s for the dominant hand (for details, see our earlier study [14]). The test-retest precision has been shown to be high for peak and average values [24]. Mean values of dominant and non-dominant sides are presented.

##### Aerobic fitness test

The subjects were given a submaximal cycle ergometer aerobic fitness test to determine their MaxVO_2_ [25].

##### Lower extremity muscle performance

Lower extremity muscle performance was measured using the timed-stands test (TST) (i.e., the number of times the subjects stand up and sit down from a standard chair for 30 s [26,27]).

### 2.4. Questionnaire

All subjects answered a questionnaire covering pain, health, distress, disability, background data, and psychological aspects.

#### 2.4.1. Pain aspects

Patients reported the duration of FM (years). Global pain intensity the previous seven days was reported using a numeric rating scale (NRS) (0 = no pain and 10 = worst possible pain).

Pain Sensitivity Questionnaire (PSQ): Pain sensitivity was assessed using the Pain Sensitivity Questionnaire (PSQ). The PSQ consists of 17 items that describe daily life situations. For each situation, the subjects rate how intense the pain (not aversiveness or distress) would be for them (0 = not painful at all and 10 = worst pain imaginable) [28]. Of the 17 situations, 14 are normally considered painful (e.g., cold, sharp, and blunt pain) and the other three are not (e.g., taking a warm shower). The mean of the 14 painful items was calculated. The PSQ was translated into Swedish using an iterative forward-backward process [29]. Although the Swedish version of the PSQ has not been psychometrically evaluated, translations to other languages have shown satisfactory statistical properties [30,31,32,33].

#### 2.4.2. Psychological distress

Hospital Anxiety and Depression Scale (HADS): The Hospital Anxiety and Depression Scale (HADS) is used frequently in clinical practice and research because of its good psychometric characteristics [34,35]. HADS includes two subscales: HADS-depression and HADS-anxiety. Both these subscales have seven items with a scoring range between 0 and 21. A lower score indicates a lower possibility of anxiety or depression. In this study, the total score (i.e., the sum of the two subscales) was used to capture psychological distress (0–42) [36].

Pain Catastrophizing Scale (PCS): The Pain Catastrophizing Scale (PCS) includes 13 items with five answering alternatives. The PCS captures different dimensions of catastrophizing such as rumination, magnification, and helplessness [37,38]. In the present study, the total PCS (PCS-total) was used; the maximum score was 52.

Insomnia Severity Index: The Insomnia Severity Index (ISI), which has good internal consistency, captures the severity and impact of insomnia symptoms [39,40]. The seven items of the ISI are rated on a five-point Likert scale (0–4) and a total score is calculated (max = 28).

#### 2.4.3. Disability

Pain Disability Index (PDI): The Pain Disability Index (PDI), which measures the impact that pain has on a person’s ability to participate in essential life activities [41,42], consists of seven items and every item is rated on a 10-point scale. The items assess the perception of the specific impact of pain on disability that may preclude normal or desired performance of a wide range of functions such as sex, work, daily activities, family and social activities, and life support (sleeping, breathing, and eating). The ratings of these seven items are summed; that is, the PDI score ranges between 0 and 70.

#### 2.4.4. Health aspects

The European Quality of Life Instrument: The European Quality of Life Instrument (EQ-5D), which captures a patient’s perceived state of health [43,44], consists of two parts. The first part is an index based upon five dimensions. In this study, however, we only used the second part: the self-estimation of today’s health according to a 100-point thermometer-like scale (EQ-VAS) with defined end points (high values indicate good health and low values indicate bad health).

### 2.5. Microdialysis and Sample Preparation

Microdialysis (MD) mimics a capillary blood vessel by perfusing a thin dialysis tube implanted in the tissue with a physiological saline solution. By simple diffusion, molecules cross the dialysis membrane along the concentration gradient and the obtained dialysate can be analyzed chemically. MD allows for continuous sampling of compounds in the interstitial space of muscle, where the nociceptors are located, and provides information of local biochemical changes before such compounds are diluted and cleared by the circulatory system. The subjects were instructed not to perform strenuous exercise two days before the study and not to drink any beverages with caffeine nor smoke the day of the study. In addition, they were asked not to take paracetamol-medication two days before and Nonsteroidal anti-inflammatory drug (NSAID) medication one week before the MD sessions. All participants indicated that they had followed the instructions. The skin and the subcutaneous tissues above the entry point of the catheter were anesthetized with a local injection (0.5 mL) of lidocaine (Xylocain_®_ 20 mg/mL, AstraZeneca, Södertälje, Sweden) without adrenaline. Care was taken not to anesthetize the underlying muscle. A commercially available MD catheter (CMA 60; CMA Microdialysis AB, Solna, Sweden) was inserted parallel to the muscle fibers of the trapezius muscle and the erector spinae muscle. This catheter had a cut-off of 20 kDa, a membrane length of 30 mm, and a diameter of 0.5 mm. Typically, a brief involuntary contraction and change of resistance were perceived when the tip of the insertion needle of the catheter entered the fascia and the muscle. Before the insertion of the catheter, ultrasound measurements determined the distance between the skin and the muscle. In the trapezius, the catheter was inserted in the middle third of the upper part of the trapezius muscle lateral to medial. Using the landmarks presented by the project Surface EMG for non-invasive assessment of muscles (SENIAM), we identified the midpoint of the line between the spine of the seventh cervical vertebra and the acromion. This point was defined as the midpoint of the descending trapezius [45]. For the erector spinae (ES) muscle, the crista iliaca was identified and the catheter was inserted 2–3 cm above this level at the most prominent part of the erector spine in the caudal direction. The catheters were perfused with a high-precision syringe pump (CMA 107; CMA/Microdialysis AB, Stockholm, Sweden) at a rate of 5 μL/min with a Ringer acetate solution (Fresenius Kabi AB, Uppsala, Sweden) containing 3-mM glucose, 0.5-mM lactate, and 3.0-µM [^14^C]-lactate (specific activity: 5.81 GBq/mmol, GE Healthcare, Buckinghamshire, UK). This procedure was performed according to the internal reference method [46]. Furthermore, nutritive muscle blood flow was estimated by the MD ethanol technique [47] using ^3^H_2_O instead of ethanol [48]. In addition, 0.3-µl/mL ^3^H_2_O (specific activity: 37 MBq/g; PerkinElmer Life Sciences, Boston, MA, USA) was added to the perfusate. The ratio of ^3^H_2_O in the dialysate and the perfusate (the outflow-to-inflow ratio) varies inversely with the local blood flow in the tissue [47,48]. In the present study, this ratio was inverted to indicate the blood flow and facilitate the interpretations.

Samples from both catheters were obtained every 20 min for the 220 min of testing and the samples were kept on ice throughout the MD experiment. The samples were then stored as aliquots at −70 °C until analysis. All vials were weighed before the experiment and after each 20-min interval to confirm that sampling and fluid recovery (FR) were working according to the perfusion rate set. Vials with visible signs of hemolysis were discarded.

Immediately after the insertion of catheters, participants rested comfortably in an armchair for 120 min (i.e., the trauma period) to allow the tissue to recover from possible changes in the interstitial environment induced. After the trauma period, participants continued to rest for 20 min, the baseline period (denoted 140 min). The baseline period was followed by a 20-min period of standardized repetitive low-force exercise of the neck-shoulder muscles sitting in a chair performed on a pegboard (denoted 160 min). The experiment ended with a recovery period of 60 min, during which participants rested in the armchair.

During the MD session, the subjects rated their pain intensity in the trapezius and erector spinae taken together from the most painful side (FM) or the dominant side (CON) every 20 min using an NRS (see above).

Analyses of metabolites: The dialysates gathered from each time point were thawed and centrifugalized. Next, the dialysates were analyzed for the interstitial concentrations of pyruvate, lactate, glutamate, glycerol, and glucose with a ISCUSS^flex^ Analyzer (CMA Microdialysis, Solna, Sweden; standard range). The detection intervals used were as follows: 0.1–12 mmol L^−1^ for lactate; 10–1500 μmol L^−1^ for pyruvate; 1.0–150 μmol L^−1^ for glutamate; 0.1–25 mmol L^−1^ for glucose; and 10–1500 mmol L^−1^ for glycerol.

Relative recovery (RR) measurements: A 5-µL dialysate or perfusate was pipetted into a counting vial containing 3-mL scintillation fluid (High-flash Point, Universal LSC-Cocktail, ULTIMA GOLD™, PerkinElmer, Inc., MA, USA) and vortexed. β-counting was performed using a liquid scintillation counter (Beckman LS 6000TA, Beckman instruments, Inc., Fullerton, CA, USA). RR was calculated as (dpm_p_ – dpm_d_)/dpm_p_, where dpm_p_ and dpm_d_ are disintegrations per minute in the perfusate and the dialysate, respectively.

Variables from MD: The MD investigations revealed blood flow, pain intensity, concentrations of the five metabolites at baseline (140 min) and after working the trapezius for 20 min (160 min), the mean value of the time points from baseline to the end of the recovery (140–220 min), and the difference between the 160 min and 140 min registrations/concentrations. In addition, pain intensity was recorded for each catheter insertion.

### 2.6. Magnetic Resonance Spectroscopy of Erector Spinae

^31^P-MRS: ^31^P-MRS of erector spinae was acquired with a Philips Ingenia 3 Tesla MR-scanner. Spectroscopic measurements were performed using a manufacturer-provided ^31^P transmit-receive surface coil with a diameter of 14 cm. Spectra were acquired using a 42 °C block pulse for the spin excitation, 15 s repetition time (TR), 153.4 μs echo time (TE), 3 kHz bandwidth, 2048 datapoints, 16 averages, and two dummy scans. 

A custom-built platform allowed the subjects to be examined in a supine head-first position while having both the MRS-coil as well as a cylindrical (r = 50 mm, h = 80 mm) external reference filled with 85 mM dimethyl methylphosphonate (DMMP) placed underneath the subject’s spinal erectors (Figure 1). ^31^P-MRS Processing: For post-processing of the ^31^P-MRS data, jMRUI [49] was used with the AMARES algorithm in combination with prior knowledge [50] for relative quantification of the resonances. Phosphocreatine (PCr) was used as a chemical-shift reference assigned to the standard chemical shift of −2.35 ppm, and other assignments were obtained from the literature as previously described [14]. Assignments are shown for the FM and CON groups in Figure 2.

In short, phosphomonoesters (PME) represented a combined signal from phosphoethanolamine (PEth) and phosphocholine (PCho); inorganic phosphate (P_i_) and PCr were all defined as singlets in jMRUI. The phosphodiester (PDE) resonance represented glycerophosphoetanolamine (GPEth) and glycerophosphocholine (GPCho). In addition, a resonance corresponding to NAD(H) was also observed. Finally, the nucleotide triphosphate (NTP-Mg or NTP for short, mainly composed of ATP saturated with Mg^2+^ [Mg-ATP]) resonances were assigned and interpreted as α-,β-, and γ-NTP-Mg as previously reported [14]. pH was estimated in the spectra using the modified Henderson–Hasselbach equation to assess the chemical shift between P_i_ and PCr (where pK_A_ = 6.77, δ_HA_ = 3.23 ppm, and δ_A_ = 5.70 ppm) as described previously [51] using the built in functionality in jMRUI.

^31^P-MRS Signal Processing: To correct for relaxation differences of the resonances, each quantized resonance was corrected according to Equation (1):(1)$$\begin{array}{cc} correction\;factor=\frac{1}{1-Exp\left(\frac{-TR}{T_{1}}\right)}\frac{1}{Exp\left(\frac{-TE}{T_{2}}\right)}, & \end{array}$$
where TR and TE are specific to the ^31^P-MRS acquisition protocol and T_1_ and T_2_ are relaxation times of the resonances being corrected. The relaxation times were obtained from the literature [52], assuming β-NTP and PME have similar T_2_ characteristics as α-NTP and PDE, respectively.

Concentrations of NTP, PCr, and P_i_ were estimated by defining a corrective factor based on the external DMMP reference as described previously [14]. During the study, the scanner required a major software upgrade and the ^31^P-coil, which was broken during the data collection phase, resulted in an increased signal-to-noise ratio (SNR) that had to be dealt with. For that reason, a correction factor for each set of software version and coil was implemented. Appendix A shows the raw SNR of the external DMMP reference as well as the PCr for each subject before and after applying the correction for the upgraded scanner software and the new coil. In addition, ratios of the phosphorous metabolites were also calculated. Hence, the following spectroscopy variables were determined: PCr (mM), P_i_ (mM), NTP (mainly ATP-Mg; in the following labelled ATP) (mM), pH, and the ratios ATP/P_tot_, PCr/P_tot_, P_i_/P_tot_, ATP/PCr, and Pi/PCr.

### 2.7. Statistics

The statistics were performed using the statistical packages IBM SPSS Statistics (version 24.0; IBM Corporation, Route 100 Somers, New York, USA) and SIMCA-P+ (version 15.0; Sartorius Stedim Biotech, Umeå, Sweden). A *P*-value < 0.05 was considered statistically significant. Text and tables report the mean value ± one standard deviation (± 1 SD) of continuous variables, and percentages (%) are reported for categorical variables. To compare groups, we used the Student’s t-test for un-paired observations and the Chi square test for proportions. Previous studies have discussed the necessity of using advanced multivariate analyses (MVDA) when accounting for system-wide aspects including missing data and multicollinearity problems [53]. Using SIMCA-P+, we applied advanced principal component analysis (PCA) to determine multivariate outliers and orthogonal partial least square regressions (OPLS) to determine multivariate associations. These techniques do not require normal distributions of the included variables [54]. SIMCA-P+ uses the non-linear iterative partial least squares (NIPALS) algorithm to handle missing data: max 50% missing data for variables/scales and max 50% missing data for subjects. In the present study, PCA was used to determine multivariate outliers. PCA extracts and displays systematic variation in the data matrix (i.e., a kind of multivariate correlation analysis). If data were skewed, variables were log transformed before the statistical analyses. A cross validation technique was used to identify nontrivial components (p). Variable loadings on the same component p were correlated, and variables with high loadings but opposing signs were negatively correlated. Variables with high absolute loadings were considered significant. Per definition, the obtained components are not correlated and are arranged in decreasing order with respect to explained variation. R^2^ describes the goodness of fit, the fraction of the sum of squares of all the variables explained by a principal component [55]. Q^2^ describes the goodness of prediction, the fraction of the total variation of the variables that can be predicted using principal component cross validation methods [55]. Outliers were identified using two methods: score plots in combination with Hotelling’s T^2^ and distance to model in X-space. In the present study, one strong multivariate outlier (a control subject–study ID 115) was identified and excluded from the multivariate analyses.

OPLS-discriminant analysis (OPLS-DA) was made for multivariate group analysis. OPLS was used for the multivariate regression analyses of pain intensity, PPTs over trapezius and erector spinae, physical tests and blood flow using the data obtained from the ^31^P-MRS, and MD examinations (except the pain intensity variables from the MD examination). The variable influence on projection (VIP) indicates the relevance of each X-variable pooled over all dimensions and Y-variables, the group of variables that best explains Y [55]. VIP > 1.0 was considered significant if VIP had a 95% jack-knife uncertainty confidence interval non-equal to zero. P(corr) was used to note the direction of the relationship (positive or negative) (i.e., the loading of each variable was scaled as a correlation coefficient and therefore standardized the range from −1 to + 1 [54]). P(corr) is stable during iterative variable selection and comparable between models. An absolute p(corr) of ≥ 0.50 was considered statistically significant [54]. Thus, a variable/regressor was considered statistically significant when VIP > 1.0 and absolute p(corr) ≥ 0.50. A regression model will be obtained including one or several components (the first is always the predictive component) if certain predefined criteria are fulfilled. The validity of the model is estimated using cross validation. Hence, for each regression, we report R^2^, Q^2^, and the *P*-value of a cross-validated analysis of variance (CV-ANOVA). The OPLS analysis was made in two steps. In the first step, all variables were included in the analysis. In the second step, the variables with VIP ≥ 1.0 were used in a new OPLS regression. The results of the second (final) regression in the text and tables are given as R^2^, Q^2^, and *P*-value of CV-ANOVA. The tables present the regressors with VIP > 1.0 and absolute p(corr) ≥ 0.50.

## 3. Results

### 3.1. Data from Questionnaires and Clinical Examinations

Significant group differences were found in disability, catastrophizing, insomnia severity, psychological distress, perceived pain sensitivity, and health-related quality of life (Table 1).

In addition, significant differences were found in blood pressure and anthropometric variables (Table 2): FM had somewhat higher blood pressures and higher BMI. In FM, the BMI classes were 30.3% normal weight, 27.3% overweight, and 42.4% obese/severe obese; in CON, the BMI classes were 71.0% normal weight, 22.6% overweight, and 6.5% obese/severe obese (Chi^2^ = 13.7, df = 1; *p* < 0.001). The measures of pain sensitivity (i.e., hyperalgesia/allodynia) also showed significant differences; in FM, PPT was lower and number of tender points higher. TST, aerobic capacity, and grip force aspects were significantly lower in FM (Table 2).

### 3.2. Microdialysis (MD)

Pain intensity was significantly higher in FM throughout MD (Table 3), whereas CON had very low levels of pain. Mean blood flow (as well as blood flow at 140 and 160 min) of the erector spinae was significantly lower in FM (Table 3); no significant differences were noted for the trapezius (Table 3). At baseline (140 min), both muscles had significantly higher pyruvate level in FM than in CON (Table 3). In addition, FM had significantly higher mean pyruvate of the trapezius than CON; a similar but non-significant trend was noted for erector spinae (*p* = 0.079). No significant group differences were found for both muscles regarding glucose, lactate, glycerol, and glutamate (Table 3).

### 3.3. Spectroscopy (^31^P-MRS)

Significantly lower absolute concentrations of ATP and PCr were found in FM, but concentrations of Pi and pH showed no significant group differences (Table 4). In FM, PCr/P_tot_ was significantly lower and Pi/PCr was significantly higher; the other ratios did not differ (Table 4).

### 3.4. Regression Analyses

The regression analyses investigated whether it was possible to regress group membership (FM or CON), pain intensity and PPT variables, blood flow in the erector spinae, and physical tests using the variables shown in Table 3 and Table 4 (except the pain intensity variables from the MD (Table 3)).

#### 3.4.1. Group Membership

In the multivariate context, blood flow aspects of the erector spinae together with PCr, ATP, and two spectroscopy ratios (PCr/P_tot_ and Pi/PCr) were significant regressors of the group differentiation (FM vs. CON) (Table 5). In addition, group differentiation was significantly influenced by pyruvate levels and blood flow of the trapezius.

#### 3.4.2. Pain Intensity in FM

Three spectroscopy ratios (Pi/PCr positively, Pi/P_tot_ positively, and PCr/P_tot_ negatively) as well as blood flow aspects of both muscles (negatively) were in the multivariate context strongest associated with pain intensity (Table 6). Additionally, pyruvate levels of erector spinae contributed to the variations in pain intensity. More than 50% of the variation (i.e. R^2^ = 0.52) in pain intensity was explained of the variables displayed in Table 6.

#### 3.4.3. Pressure Pain Thresholds (PPT) for Trapezius and Erector Spinae

Since FM and CON differed markedly in PPTs (Table 2), an OPLS regression of all subjects will mainly reflect the group differentiation (cf. Table 5). The regressions were made separately in each group. The OPLS of PPT over erector spinae included both spectroscopy and MD data while the analysis of PPT over the trapezius used MD data since spectroscopy not was done for this muscle.

It was not possible to significantly regress PPT in CON in either of the two muscles.

##### PPT of trapezius

In FM, increases in glutamate, lactate, glycerol, and pyruvate from 140 min to 160 min correlated positively with PPT (Table 7). Glutamate level at 160 min and blood flow aspects of trapezius also contributed and correlated positively with PPT.

##### PPT of Erector spinae

In FM Pi/PCr, Pi/P_tot_, two glycerol aspects and pyruvate at 140 min were the most important regressors of PPT (Table 8).

#### 3.4.4. Blood Flow in Erector Spinae

In this analysis, spectroscopy data and baseline MD data were used as regressors of blood flow at baseline for erector spinae. We obtained significant regressions for all subjects taken together and, in each group, separately (Table 9). Five significant regressors were obtained for each regression and four of these were the same although with different relative importance within each regression: ATP (positively correlated); Pi/PCr (negatively), Pi/P_tot_ (negatively); and PCr/Ptot (positively). In all subjects and in FM, but not in CON, PCr was a significant regressor of blood flow.

#### 3.4.5. Physical Tests

Grip strength aspects: It was impossible to obtain significant regressions in the groups separately. In all subjects (i.e., as a group), the strongest associations with grip strength were pyruvate variables from both muscles (negatively) and blood flow of thee erector spinae (positively) (Appendix A).

Lower extremity muscle performance (TST): In all subjects (i.e., as a group), the strongest associations with TST were Pi/PCr and Pi/P_tot_ (both negatively), mean pyruvate levels of the two muscles (negatively), and baseline blood flow of erector spinae (positively) (Appendix A). In FM, the strongest regressors in the significant regression of TST were pyruvate, glutamate, and glucose aspects (all negatively) (Appendix A). In CON, no significant regressions of TST were obtained.

Aerobic fitness: No significant regressions for aerobic fitness were obtained for all subjects (as a group) or the CON. In FM, however, aerobic fitness showed the strongest associations with glycerol aspects (negatively and positively), pyruvate (negatively), and lactate (negatively) levels in the trapezius and to some extent the erector spinae (Appendix A).

## 4. Discussion 

(Original studies using other FM criteria than ACR 1990 criteria have been indicated.)

### 4.1. Major Results

FM was clearly associated with higher levels of pyruvate and lower levels of ATP and PCr, a finding that suggests muscle mitochondrial dysfunctions in FM. The multivariate analysis of the groups (FM vs. CON) (Table 5) showed that blood flow of the erector spine together with the spectroscopy variables PCR, ATP, PCr/P_tot_, and Pi/PCr were more important for group differentiation than the other variables that exhibited significant group differences. The multivariate associations between metabolic variables and pain aspects (intensity and PPTs) indicated that peripheral factors contribute to the perception of pain in FM.

### 4.2. Significantly Higher Levels of Pyruvate in FM

The breakdown of glucose and glycogen in the cytosol results in pyruvate and lactate. Increased pyruvate was found in both muscles of FM, a finding that agrees with two other MD studies of trapezius that reported increased levels of pyruvate and/or lactate [9,10]. Both substances were also significantly increased in the vastus lateralis in a third FM cohort [11]. Based on the three muscles studied in these four FM cohorts, FM seems to be associated with increases in key products of glycolysis (i.e., pyruvate and possibly lactate). The increased pyruvate level in the trapezius cannot be explained by insufficient oxygen supply as no blood flow alteration was found in this cohort or in the two other FM cohorts [10,11]. In addition, the analyses of blood flow in the erector spinae supports this interpretation as no association with pyruvate levels exists (Table 9).

The mitochondrial pyruvate carrier (MPC) is responsible for the transportation of pyruvate into mitochondria and its use in the production of ATP [7]. Pyruvate can be reduced to lactate and vice versa in the cytosol by the bi-directional enzyme lactate dehydrogenase (LDH) [7,56]. Challenging traditional views regarding glycolysis, it is now evident that lactate is produced both during anaerobic and aerobic conditions and can be metabolized in the same cell or transported to other cells via lactate shuttles [57,58]. Under aerobic conditions, glycolysis produces lactate, which muscle mitochondria use to produce ATP [57,58]. Lactate is transported via monocarboxylate transporters into the mitochondrion and converted to pyruvate. This pyruvate, via pyruvate dehydrogenase, is converted into Acetyl-CoA before it enters the tricarboxylic acid cycle (TCA) and ultimately contributes to oxidative phosphorylation (OXPHOS) and the electron transport chain (ECT), and therefore the production of ATP [57,58,59,60].

### 4.3. Lower Absolute Concentrations of PCr and ATP of Erector Spinae in FM

ATP is critical for muscle contractile activity and the demand can increase more than 100-fold during work [61,62]. To provide energy for muscular contraction, ATP splits into ADP, Pi, and H^+^ ions by ATPase enzyme [63]. PCr acts as an intermediate energy buffer and “shuttles” energy-rich phosphate present in ATP from the mitochondria to the myofibrils [64]. As intramuscular stores of ATP are small, free fatty acids and glucose/glycogen pathways are also activated to produce ATP [62]. Two earlier studies of FM determined the absolute concentrations of PCr and ATP [14,15] and both found significantly lower levels of these in the quadriceps muscle. These results were confirmed for the erector spinae in this study (Table 4). Lower ATP levels seem to characterize FM as low ATP levels have also been found in skin, plasma, platelets, neuronal cells, and peripheral blood mononuclear cells of FM patients [65,66,67]. The decreased ATP and PCr levels were parallel in FM as the ATP/PCr ratio was not different between the two groups (Table 4). This finding agrees with the two earlier ^31^P-MRS studies based on our calculations of their data [14,15]. The ratios involving PCr and phosphorus (i.e., PCr/P_tot_ and Pi/PCr) both showed significant group differences, which is mainly due to the significantly lower PCr in FM as the concentration of Pi and the Pi/P_tot_ did not differ (Table 4). Hence, these two ratios indicate alterations in the PCr-phosphorus balance. The lower ATP production in FM subsequently results in lower PCr concentration, whereas the P_i_ concentration is not affected and therefore the P_i_/PCr ratio increased in FM, ultimately decreasing PCr/P_tot_ (Table 4). It was evident that these two ratios were also important for pain aspects and blood flow.

### 4.4. What is the Explanation for Metabolic and Blood Flow Alterations?

Mitochondrial dysfunction could explain the increased pyruvate levels and lower absolute concentrations of ATP and PCr in FM. The aerobic capacity of muscles is largely governed by the number of mitochondria and their enzymes [68]. Mitochondria are highly dynamic organelles and a proper balance between fusion and fission of mitochondria is important for integrity and function including reactive oxygen species (ROS) generation, apoptosis regulation, and energy metabolism [8,69,70]. Our explanation agrees with previous FM studies that found trapezius muscle fibers had mitochondrial alterations [71,72]. Moreover, different muscle alterations have been reported for quadriceps [73,74,75]. Mitochondrial alterations are not specific for FM and have also been found in muscles that exhibit localized chronic pain [76,77]. Mitochondrial conditions are divided into primary and secondary mitochondrial diseases [78]. The latter—possibly present in FM—can be acquired secondary to adverse environmental effects that result in oxidative stress. Peripheral metabolic alterations have also been observed in other FM studies. A proteomic study of the trapezius muscle of FM found significant alterations in metabolic proteins involved in ATP production and in glycolysis and glucogenesis such as lower concentrations of creatine kinase, a situation that could contribute to lower concentrations of PCr [79]. In addition, plasma metabolomic studies have found that FM differed from healthy subjects in energy, lipid, and amino acid metabolites [80]. Altered metabolic profile in urine has also been reported in patients with FM [81,82]. Larger prospective studies of other peripheral molecules suggest that FM is associated with peripheral biochemical alterations in muscle and in plasma [83,84,85,86,87,88,89] (Han et al. used ACR 2010 criteria).

The increased pyruvate levels may reflect insufficient transportation of pyruvate into the mitochondria or insufficient reduction of pyruvate to lactate and/or insufficient transport of lactate into the mitochondria. Both these conditions would result in a decrease of ATP and PCr. In addition, increased pyruvate levels may be the result of an inflammatory environment that switches metabolism from OXPHOS to glycolysis, a condition that would prevent pyruvate from being transported into mitochondria. In this scenario, LDH in the cytosol converts two pyruvate molecules back into lactate, resulting in only two ATP molecules rather than the 36 molecules generated by OXPHOS [90], a decrease in energy production.

Significant associations were found mainly between the spectroscopy variables and blood flow in the erector spinae at baseline (Table 9). As in other studies, we found that blood flow was positively associated with ATP; intraluminal ATP is a vasodilator that can lead to both activation of vasoactive substances (e.g., nitric oxide, NO) and prostacyclin by activation of inwardly rectifying potassium channels [91]. Moreover, PCr concentration and the balance between PCr and phosphorus were multivariately associated with blood flow (Table 9). At this time, causality is unclear and cannot be determined in this study. In addition, other factors such as increased sympathetic tone may contribute to lower blood flow in FM [92]. The unaltered blood flow in trapezius muscle agrees with earlier studies of this muscle in FM [10,11]. A conflicting situation exists regarding muscle blood flow in FM [93]. The different situations in the two muscles may indicate that FM muscles have different predilections for blood flow changes and/or temporal aspects (i.e., the pain condition is first initiated in the neck-shoulder muscles).

### 4.5. What Is the Reason for Mitochondrial Dysfunction?

The observation of reduced physical capacity in FM (Table 2) agrees with other studies [94,95] (Umay et al. used ACR 2013 criteria). The mitochondrial density and capacity increases as the result of exercise [69,70]. Aging, immobilization, and long-term physical inactivity are associated with negative changes in content and function of mitochondria [95,96]. Muscle disuse/inactivity causes large loss in muscle mass that is preceded by changes in mitochondrial content [70]. According to reviews, aerobic and muscle strengthening exercise interventions reduce pain and improve general well-being in patients with FM [97,98]. Exercise is associated with upregulations of mitochondrial protein synthesis and therefore the capacity for ATP generation, oxygen delivery, and antioxidant capacity [99]. It is not known if such effects will also be achieved in FM and if they are associated with improvements in the clinical presentation. An exercise intervention of FM found that increased pyruvate levels in vastus lateralis were normalized after a 16-week exercise intervention [11]. Therefore, at least some metabolic alterations in FM can be addressed successfully using exercise. Mitochondrial function has also been linked to insulin sensitivity: individuals whose ATP production increased as a result of exercise also had enhanced insulin sensitivity [100]. FM had higher BMI than CON (Table 2). Furthermore, physical exercise had fewer positive effects on mitochondrial function in obese subjects than in lean subjects [100]. Deconditioning is associated with high sympathetic tone and low para-sympathetic tone [92]. Muscle strength interventions may reduce sympathetic hyperactivity and abnormal vagal balance [94].

However, mitochondrial disturbances may not be directly linked to physical activity level. Mitochondrial dysfunction in the skin may indicate that factors—possibly interacting with each other—other than inactivity are responsible [65,66]. Aging and obesity are both associated with mitochondrial dysfunction [76,96]. Partly in agreement with these observations, we found that aside from group (FM or CON), BMI—but not age—was a significant regressor of PCr, ATP, and puruvate of trapezius according to OPLS regressions. FM had more insomnia than CON (Table 1); interestingly, bidirectional associations exist between circadian rhythms and mitochondrial activity [90]. In addition, common comorbidities in FM may be responsible; diabetes, cardiovascular disease, and neurogenerative disorders are associated with secondary mitochondrial dysfunctions [78]. Altered activation patterns of the FM muscle [101,102,103,104] may indicate muscular over-activity and therefore alterations in pyruvate and/or lactate levels [105,106]. Mitochondria dysfunction may also be linked to the pain per se as discussed below.

### 4.6. Multivariate Associations between Pain Aspects and Spectroscopy and MD Variables

High pain intensity is a considerable burden for patients with FM. At the group level, pain intensity was above six on the NRS (Table 1). Pain intensity correlated with the metabolic situation and blood flow aspects; >50% of the variation in pain intensity was explained by these factors (Table 6). Specifically, PCr/P_tot_ and Pi/PCr together with blood flow aspects of both muscles, where the Pi/P_tot_ and pyruvate aspects were important regressors of pain intensity (Table 6). A proteomic study of mainly patients with FM found that 12 muscle proteins showed strong multivariate associations with pain intensity [107]; proteins related to ATP synthesis were among the important proteins [107]. In addition, certain plasma proteins showed strong multivariate correlations with pain intensity in two cohorts of mainly FM [87,108]. In contrast, an inflammatory panel of 92 cytokines/chemokines from plasma were not correlated with pain intensity [109]. In the exercise intervention above-mentioned, the degree of normalization in pyruvate was significantly associated with decreases in pain intensity although the association was weak [11]. Thus, exercise interventions normalizing muscle metabolic situation may affect pain intensity.

As expected, the PPT registrations confirmed the presence of generalized hyperalgesia/allodynia in FM (Table 2) [110,111,112,113] (different criteria used in these references). PPT measures when an acute mechanical stimulus becomes painful; in healthy subjects, the stimuli activates both the peripheral nociceptors and the CNS. In FM, peripheral metabolism and blood flow alterations influenced PPT. It was not possible to regress PPT in CON because factors other than those in FM are important for PPT; such group differences were also found in a plasma proteomic study [114]. PPT over trapezius (spectroscopy not made) was positively associated with the differences in blood flow and in concentrations of glutamate, lactate, glycerol, and pyruvate (Table 7). Therefore, more prominent metabolic changes during highly repetitive work were associated with higher PPT in FM. Pi/PCr, Pi/P_tot_, two glycerol aspects and pyruvate at 140 min were the most important regressors of PPT over erector spinae (Table 8). Associations between peripheral biochemical molecules and PPT in FM have been reported previously where muscle (including proteins related to ATP synthesis) and plasma proteins showed strong multivariate associations with PPT [87,107,114]. The acute pressure used to elicit PPT is a peripheral input, which in FM partly reflects the metabolic status in muscle tissue. As only approximately one-third of the variation in PPT was explained, other peripheral and/or central factors—not mutually exclusive—contribute in FM (e.g., a continuous nociceptive input, peripheral nociceptor sensitization, secondary hyperalgesia in the primary pain region, and a generalized state of hypersensitivity such as central sensitization including impaired descending control of nociception).

As chronic pain is common in patients with different mitochondrial diseases [115], it would be helpful to know whether metabolic alterations directly interact with nociception and pain in FM or whether these metabolic alterations reflect other mitochondrial mechanisms more directly associated with nociception and pain. For example, increased lactate and pyruvate levels can induce ROS, which in turn may interact with nociception [116,117,118,119]. However, we suggest that other aspects of dysfunctional mitochondria are responsible and interact with immune and nociceptive systems. Mitochondrial dysfunction can lead to inflammatory conditions [120]. Mitochondria have a vital role in the regulation of immune cells and contribute to immune responses [121]. Dysfunctional mitochondria release damage-associated molecular patterns recognized by the immune system (e.g., increased levels of mtROS or mtDNA) that trigger immune and inflammatory responses [90]. Normally, dysfunctional mitochondria that produce high mtROS are removed by mitophagy to avoid harmful inflammation effects [122]. Increased ROS levels have been reported in FM [123,124]. Excessive production of mtROS has been associated with aging, osteoarthritis, and rheumatoid arthritis [90]. Increased mtROS levels can activate inflammasomes (NLRP3) that stimulate caspase-1, which upregulates several inflammatory cytokines (e.g., IL-1β and IL-18) [120]. NLRP3 may be activated in FM [125]. IL-1β causes pain by directly acting on sensory neurons via their receptors [126] and stimulates production of additional algogenic substances [127]. A systematic review [128] was unable to confirm increased plasma levels of IL-1β (also in blood cells) and IL-18 in FM [125]. These conflicting results for IL-1β may be due to IL-1β’s anti-inflammatory and pro-inflammatory properties [129]. Mitochondria can also modulate innate immunity via nuclear factor kappa B (NF-кB) and interferon regulatory factor I pathways with subsequent activation of TNF-α, IL-1, IL-6, IL-8, IFNβ, and IFNλ1 [8,90]. Other dysfunctions of mitochondria (e.g. calcium overload, increased apoptosis, decreased mitophagy, and oxidized mtDNA) can also activate inflammatory pathways [120,122].

### 4.7. Strengths and Limitations

We used the ACR 1990 classification criteria for FM to simplify comparisons with earlier studies. In future studies, both ACR 1990 and newer criteria should be used to optimize comparisons with other studies. During the study, the MR scanner software was upgraded, and the ^31^P-coil was replaced, changes that have the potential of introducing limitations. However, as we ensured that the FM and CON subjects were scanned intermittently, these changes are not likely to affect the outcome. In addition, there was no significant difference in the proportion of the subjects who were examined with different versions and coils (Supplementary data 1). Another strength is that we used multivariate techniques. Typically, FM studies are hypothesis-driven studies that focus on a few molecules. To achieve a mechanistic understanding of the biological factors maintaining pain conditions, it is necessary to understand the activated molecular mechanisms from a broader system biology perspective [130,131], a conclusion that influenced how we analyzed the data. We combined traditional statistical methods with advanced multivariate methods to handle intercorrelated variables.

## 5. Conclusions

We found significant metabolic and blood flow alterations in the muscles of FM. The results may indicate muscle mitochondrial dysfunctions in FM. Although it is unclear why muscle mitochondrial dysfunctions are found in FM, inactivity, obesity, aging, and pain per se may be involved. The obtained significant regressions of pain intensity and PPT in FM agree with other studies that report significant multivariate associations between peripheral biological factors and pain aspects. We believe that our results may be important for understanding the pathophysiological mechanisms in FM and ultimately contribute to developing effective treatments for FM.

## Figures and Tables

**Figure 1 jcm-09-03527-f001:**
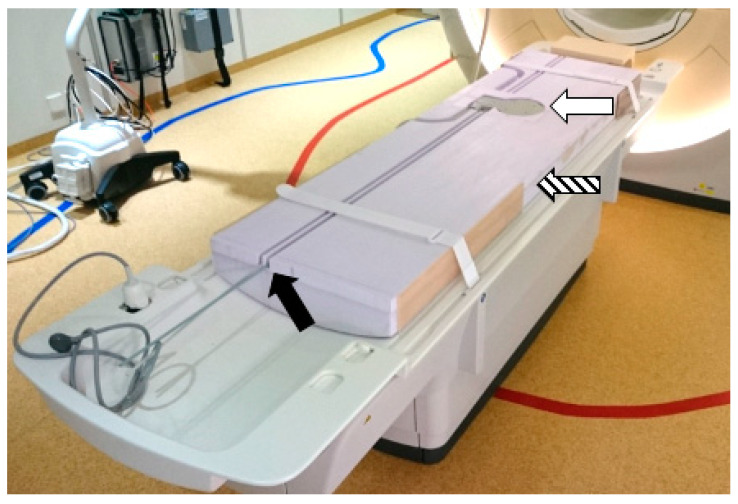
Experimental setup for measuring phosphorus-31 magnetic resonance spectroscopy (31P MRS) at 3 T, within the spinal erectors. A custom platform was built (striped arrow) to fit on the scanner table. This platform allowed the participants to lie flat on their backs with the phosphorus transmit/receive coil lowered into the platform (white arrow). Underneath the coil, a cavity housed the external cylindrical reference. Spaces were also made for the tuning and matching control rods (black arrow).

**Figure 2 jcm-09-03527-f002:**
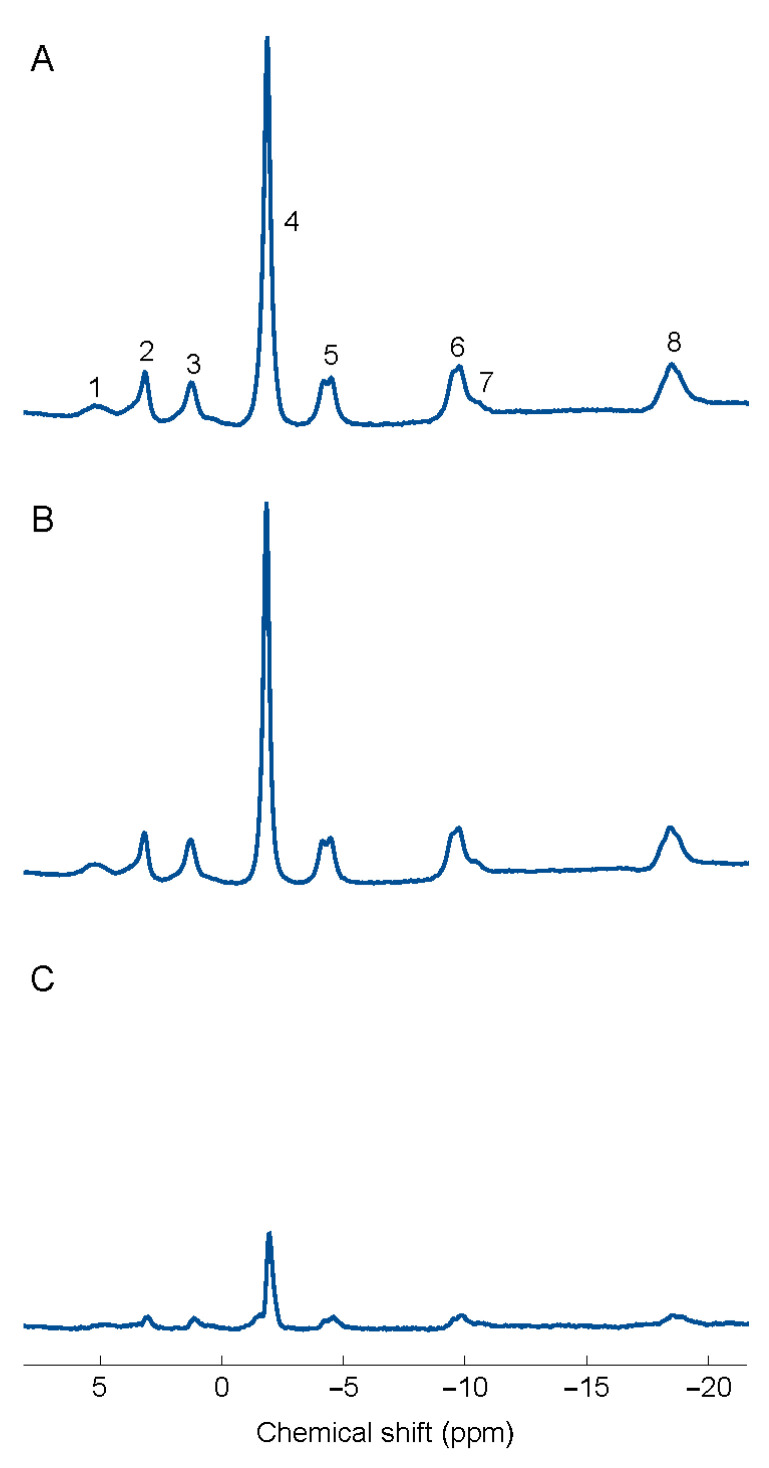
Spectra corresponding to the mean for the (**A**) control group (*n* = 30) and the corresponding (**B**) patient group with fibromyalgia (*n* = 32) in the spinal erectors at 3 T. Spectra are all normalized in Table (1) Phosphomonoesters(PME), (2) inorganic phosphate (Pi), (3) phosphodiesters (PDE), (4) phosphocreatine (PCr), (568) γ, α, and β-nucleotide triphosphate (mainly Mg-saturated ATP), and (7) NAD(H). (**C**) The difference between the two groups of subjects.

**Table 1 jcm-09-03527-t001:** Data obtained from questionnaires in controls and in patients with fibromyalgia (FM).

Group	CON			FM			Statistics
Variables	*n*	Mean	SD	*n*	Mean	SD	*p*-value
Age (years)	31	42.26	10.17	33	40.06	11.36	0.417
FM duration (years)	−	−	−	33	6.12	6.04	NA
Pain intensity global 7days	−	−	−	31	6.55	1.71	NA
HADS	31	4.23	3.59	31	13.32	6.17	< 0.001 *
PCS	31	11.45	8.97	31	19.94	10.34	0.001 *
ISI	31	4.23	4.47	31	13.58	5.96	< 0.001 *
PSQ	31	3.50	0.91	31	5.64	1.80	< 0.001 *
PDI	30	8.43	4.93	31	36.45	11.25	< 0.001 *
EQ5D-VAS	30	86.67	7.54	31	52.16	18.85	< 0.001 *

CON = controls; FM = fibromyalgia; SD = standard deviation; HADS = The Hospital Anxiety and Depression Scale; PCS = Pain Catastrophizing Scale; ISI = Insomnia Severity Index, PSQ-total = Pain Sensitivity Questionnaire-total scale; PDI = Pain Disability Index; EQ5D-VAS = Perceived Health according to EQ instrument; NA = Not Applicable. * denotes statistical significance.

**Table 2 jcm-09-03527-t002:** Variables obtained at the clinical examinations.

Group	CON			FM			Statistics
Variables	*n*	Mean	SD	*n*	Mean	SD	*p*-value
Blood pressure systolic (mm Hg)	31	113.35	8.64	33	120.88	12.58	0.007 *
Blood pressure diastolic (mm Hg)	31	75.58	8.23	33	80.55	10.07	0.035 *
Weight (kg)	31	68.57	10.72	33	81.04	18.68	0.002 *
Height (m)	31	1.70	0.06	33	1.66	0.06	0.038 *
BMI (kg/m^2^)	31	23.81	3.16	33	29.18	6.13	<0.001 *
Number of tender points	31	0.32	0.87	33	16.48	2.66	<0.001 *
Mean PPT all sites (kPa)	31	383.66	109.91	33	132.55	84.66	<0.001 *
PPT Trapezius catheter side (kPa)	31	303.74	103.01	33	109.54	69.01	<0.001 *
PPT Erector Spine catheter side (kPa)	31	432.61	147.15	33	121.79	87.91	<0.001 *
MaxVO_2_ per kg (mL/kg/min)	30	2.64	0.49	30	2.17	0.53	0.001 *
Grip force-max (N)	30	311.59	51.58	33	240.38	63.94	<0.001 *
Grip force-average (N)	30	232.77	45.26	33	162.31	54.07	<0.001 *
Grip force-endurance (N)	30	233.05	127.05	33	137.80	47.75	<0.001 *
TST	30	17.80	3.03	32	13.63	3.47	<0.001 *

CON = controls; FM = fibromyalgia; SD = standard deviation; BMI = Body Mass Index; PPT = Pressure Pain Threshold; TST = Timed-Stands Test; * denotes statistical significance.

**Table 3 jcm-09-03527-t003:** Data obtained from the microdialysis of trapezius and erector spinae muscles.

Groups	Controls			FM			Statistics
Variables	*n*	Mean	SD	*n*	Mean	SD	*p*−value
NRS pre insertion	31	0.13	0.43	33	4.82	2.89	<0.001 *
NRS 140 min	31	0.42	0.85	33	5.55	2.21	<0.001 *
NRS 160 min	31	1.74	1.84	33	7.67	1.67	<0.001 *
Difference NRS 160 min−140 min	31	1.32	1.38	33	2.12	1.75	0.047 *
Mean NRS 140−220 min	31	0.61	0.84	33	6.07	1.85	<0.001 *
Blood flow Trapezius 140 min	30	0.68	0.07	32	0.65	0.07	0.054
Blood flow Trapezius 160 min	30	0.70	0.10	32	0.67	0.11	0.249
Difference Blood flow Trapezius 160 min−140 min	30	0.02	0.05	32	0.03	0.09	0.825
Mean Blood flow Trapezius 140−220 min	30	0.69	0.07	32	0.66	0.07	0.107
Blood flow Erector spinae 140 min	26	0.54	0.19	31	0.36	0.19	0.001 *
Blood flow Erector spinae 160 min	26	0.59	0.17	31	0.39	0.19	<0.001*
Difference Blood flow Erector spinae 160 min−140 min	25	0.03	0.08	31	0.03	0.08	0.886
Mean Blood flow Erector spinae 140−220 min	27	0.55	0.18	31	0.37	0.18	<0.001 *
Glucose Trapezius 140 min (mmol L^−1^)	29	10.00	6.92	32	8.41	5.24	0.312
Glucose Trapezius 160 min (mmol L^−1^)	30	13.25	19.05	32	7.89	5.44	0.132
Difference Glucose Trapezius 160 min–140 min (mmolLl^−1^)	29	3.11	19.08	32	−0.52	4.94	0.303
Mean Glucose Trapezius 140−220 min (mmol L^−1^)	30	9.65	5.94	32	8.27	4.39	0.300
Glucose Erector spinae 140 min (mmol L^−1^)	24	8.35	6.32	31	9.12	6.33	0.657
Glucose Erector spinae 160 min (mmol L^−1^)	24	12.89	21.88	31	12.57	26.87	0.962
Difference Glucose Erector spinae 160 min−140 min (mmol L^−1^)	23	4.91	22.89	31	3.45	28.59	0.841
Mean Glucose Erector spinae 140−220 min (mmol Lv^1^)	25	9.48	6.34	31	10.55	8.81	0.613
Lactate Trapezius 140 min (mmol L^−1^)	28	2.55	1.62	32	2.71	1.67	0.709
Lactate Trapezius 160 min (mmol L^−1^)	28	4.61	7.29	32	2.88	2.05	0.234
Difference Lactate Trapezius 160 min − 140 min (mmol L^−1^)	27	2.19	6.80	32	0.17	1.90	0.145
Mean Lactate Trapezius 140−220 min (mmol L^−1^)	30	3.51	2.33	32	3.22	1.48	0.552
Lactate Erector spinae 140 min (mmol L^−1^)	23	1.86	1.04	29	2.68	3.80	0.275
Lactate Erector spinae 160 min (mmol L^−1^)	23	2.91	4.32	29	2.74	2.66	0.862
Difference Lactate Erector spinae 160 min−140 min (mmol L^−1^)	23	1.04	4.21	28	0.08	3.36	0.369
Mean Lactate Erector spinae 140–220 min (mmol L^−1^)	23	2.23	1.36	30	3.65	5.58	0.237
Pyruvate Trapezius 140 min (μmol L^−1^)	30	12.97	10.98	32	27.96	18.19	< 0.001 *
Pyruvate Trapezius 160 min (μmol L^−1^)	30	25.54	34.47	32	33.84	31.16	0.323
Difference Pyruvate Trapezius 160 min−140 min (μmol L^−1^)	30	12.57	31.74	32	5.88	24.81	0.357
Mean Pyruvate Trapezius 140−220 min (μmol L^−1^)	30	20.10	17.87	32	38.64	29.75	0.005 *
Pyruvate Erector spinae 140 min (μmol L^−1^)	25	12.37	10.97	31	26.59	30.74	0.032 *
Pyruvate Erector spinae 160 min (μmol L^−1^)	25	27.22	45.69	31	80.37	270.52	0.337
Difference Pyruvate Erector spinae 160 min−140 min (μmol L^−1^)	24	14.66	46.09	31	53.79	275.61	0.495
Mean Pyruvate Erector spinae 140−220 min (μmol L^−1^)	26	18.57	15.97	31	44.15	71.35	0.079

Glycerol Trapezius 140 min (mmol L^−1^)	30	90.37	60.04	32	95.26	52.10	0.733
Glycerol Trapezius 160 min (mmol L^−1^)	29	108.94	82.71	32	105.91	85.78	0.889
Difference Glycerol Trapezius 160 min−140 min (mmol L^−1^)	29	19.35	92.43	32	10.65	72.58	0.683
Mean Glycerol Trapezius 140−220 min (mmol L^−1^)	30	90.06	41.68	32	103.28	57.53	0.307
Glycerol Erector spinae 140 min (mmol L^−1^)	25	138.79	96.15	31	149.73	119.08	0.712
Glycerol Erector spinae 160 min (mmol L^−1^)	25	141.55	177.53	29	136.95	101.51	0.906
Difference Glycerol Erector spinae 160 min − 140 min (mmol L^−1^)	24	7.17	198.28	29	−10.62	97.96	0.673
Mean Glycerol Erector spinae 140−220 min (mmol L^−1^)	26	126.60	65.65	31	150.91	90.05	0.258
Glutamate Trapezius 140 min (mmol L^−1^)	30	49.77	31.65	32	56.06	32.50	0.444
Glutamate Trapezius 160 min (mmol L^−1^)	29	81.88	56.69	32	79.47	55.15	0.867
Difference Glutamate Trapezius 160 min−140 min (mmol L^−1^)	29	32.56	57.77	32	23.42	40.17	0.473
Mean Glutamate Trapezius 140−220 min (mmol L^−1^)	30	63.24	30.87	32	67.55	34.33	0.606
Glutamate Erector spinae 140 min (mmol L^−1^)	25	41.40	26.52	31	30.20	43.02	0.260
Glutamate Erector spinae 160 min (mmol L^−1^)	25	48.96	78.95	31	41.44	88.96	0.742
Difference Glutamate Erector spinae 160 min − 140 min (mmol L^−1^)	24	6.21	79.89	31	11.24	88.62	0.828
Mean Glutamate Erector spinae 140 − 220 min (mmol L^−1^)	26	41.79	26.23	31	33.82	39.23	0.381

NRS = Numeric rating scale for pain intensity. * denotes statistical significance.

**Table 4 jcm-09-03527-t004:** Results obtained from spectroscopy examination of the erector spinae muscle.

Group	CON			FM			Statistics
Variables	*n*	Mean	SD	*n*	Mean	SD	*p*-value
PCr (mM)	30	40.06	8.54	32	34.07	11.49	0.024 *
Pi (mM)	30	5.68	1.84	32	5.29	1.76	0.409
ATP (mM)	30	8.64	1.49	32	7.57	1.91	0.017 *
pH	30	7.03	0.03	32	7.03	0.03	0.514
Ratio ATP/P_tot_	30	0.30	0.02	32	0.30	0.02	0.897
Ratio PCr/P_tot_	30	1.37	0.06	32	1.30	0.13	0.018 *
Ratio Pi/Ptot	30	0.19	0.03	32	0.20	0.03	0.098
Ratio ATP/PCr	30	0.22	0.02	32	0.23	0.04	0.101
Ratio Pi/PCr	30	0.14	0.03	32	0.16	0.04	0.017 *

CON = controls; FM = fibromyalgia; SD = standard deviation; PCr = phosphocreatine; Pi = inorganic Phosphate; P_tot_ = total phosphorus; ATP = adenosine triphosphate. * denotes statistical significance.

**Table 5 jcm-09-03527-t005:** OPLS-DA of group membership (CON denoted 0 and FM denoted 1). Regressors used data from spectroscopy and microdialysis (except pain variables from microdialysis) (cf. Table 3 and Table 4). The regression had one predictive component. Only significant variables are shown (i.e., variables with VIP > 1.0 and absolute p(corr) ≥ 0.50).

Variables	VIP	p(corr)
Blood flow Erector spinae 140 min	2.37	−0.83
Mean Blood flow Erector spinae 140−220 min	2.27	−0.81
Blood flow Erector spinae 160 min	2.23	−0.80
PCR (mM)	1.97	−0.70
PCr/P_tot_	1.94	−0.69
ATP (mainly ATP; mM)	1.94	−0.69
Pi/PCr	1.82	0.65
Pyruvate trapezius 140 min	1.60	0.57
Mean Blood flow Trapezius 140–220 min	1.57	−0.55
ATP/PCr	1.49	0.53
Blood flow Trapezius 160 min	1.42	−0.50
R^2^	0.26	
Q^2^	0.18	
CV-ANOVA *p*-value	0.003	
*n*	61	

VIP and p(corr) are reported for each regressor (i.e., the loading of each variable scaled as a correlation coefficient and therefore standardizing the range from −1 to + 1). The sign of p(corr) indicates the direction of the correlation with the dependent variable (+ = positive correlation; − = negative correlation). Hence, a negative p(corr) indicates lower values for a certain variable in FM in the multivariate context. The four bottom rows report R^2^, Q^2^, *P*-value of the CV-ANOVA, and number of subjects included in the regression (*n*) from the second OPLS regression (see Statistics for details).

**Table 6 jcm-09-03527-t006:** OPLS of global pain intensity previous seven days in FM. Regressors used data from spectroscopy and microdialysis (except pain variables from microdialysis) (cf. Table 3 and Table 4). The regression had one predictive component. Only significant variables are shown (i.e., variables with VIP > 1.0 and absolute p(corr) ≥ 0.50).

Variables	VIP	p(corr)
Pi/PCr	1.71	0.65
Mean Blood flow Trapezius 140–220 min	1.70	−0.64
Glutamate Trapezius 160 min	1.53	−0.57
Blood flow Trapezius 140 min	1.52	−0.57
Blood flow Erector spinae 140 min	1.51	−0.57
Pi/P_tot_	1.51	0.57
PCr/P_tot_	1.51	−0.57
Difference Pyruvate Erector spinae 160 min–140 min	1.48	0.56
Pyruvate Erector spinae 160 min	1.48	0.56
Blood flow Trapezius 160 min	1.48	−0.55
Difference Glutamate Erector spinae 160 min–140 min	1.46	0.55
Mean Pyruvate Erector spinae 140–220 min	1.46	0.55
Difference Glucose Erector spinae 160 min–140 min	1.39	0.52
PCr	1.38	−0.53
Mean Blood flow Erector spinae 140–220 min	1.37	−0.51
Blood flow Erector spinae 160 min	1.35	−0.51
R^2^	0.52	
Q^2^	0.32	
CV-ANOVA *p*-value	0.006	
*n*	30	

VIP and p(corr) are reported for each regressor (i.e., the loading of each variable scaled as a correlation coefficient and therefore standardizing the range from −1 to + 1). The sign of p(corr) indicates the direction of the correlation with the dependent variable (+ = positive correlation; − = negative correlation). Hence, a positive p(corr) for a certain variable indicates a positive correlation with pain intensity in FM in the multivariate context. The four bottom rows report R^2^, Q^2^, *P*-value of the CV-ANOVA, and number of subjects included in the regression (*n*) from the second OPLS regression (see Statistics for details).

**Table 7 jcm-09-03527-t007:** OPLS of PPT over trapezius in FM. Regressors used data from microdialysis of this muscle (except pain variables from microdialysis) (cf. Table 3). Only significant variables are shown (i.e., the variables with VIP > 1.0 and absolute p(corr) ≥ 0.50).

Variables	VIP	p(corr)
Difference Glutamate Trapezius 160 min–140 min	1.84	0.78
Difference Lactate Trapezius 160 min–140 min	1.71	0.72
Glutamate Trapezius 160 min	1.51	0.64
Blood flow Trapezius 140 min	1.41	0.60
Difference Glycerol Trapezius 160 min–140 min	1.39	0.59
Mean Blood flow Trapezius 140–220 min	1.36	0.57
Difference Pyruvate Trapezius 160 min–140 min	1.20	0.51
R^2^	0.32	
Q^2^	0.19	
CV-ANOVA P-value	0.050	
*n*	30	

VIP and p(corr) are reported for each regressor (i.e., the loading of each variable scaled as a correlation coefficient and therefore standardizing the range from −1 to + 1). The sign of p(corr) indicates the direction of the correlation with the dependent variable (+ = positive correlation; − = negative correlation). Hence, a positive p(corr) for a certain variable indicates a positive correlation with PPT over trapezius in FM in the multivariate context. The four bottom rows report R^2^, Q^2^, P-value of the CV-ANOVA, and number of subjects included in the regression (*n*) from the OPLS regression (see Statistics for details).

**Table 8 jcm-09-03527-t008:** OPLS of PPT over erector spinae in FM. Regressors used data from microdialysis and spectroscopy of this muscle (except pain variables from microdialysis) (cf. Table 3 and Table 4). The regression had one predictive component. Only significant variables are shown (i.e., variables with VIP > 1.0 and absolute p(corr) ≥ 0.50).

Variables	VIP	p(corr)
Pi/PCr	1.73	−0.69
Pi/P_tot_	1.58	−0.63
Mean Glycerol Erector spinae 140–220 min	1.52	−0.60
Glycerol Erector spinae 140 min	1.51	−0.60
Pyruvate Erector spinae 140 min	1.46	−0.58
Difference Blood flow Erector spinae 160 min–140 min	1.44	−0.57
Blood flow Erector spinae 140 min	1.41	0.56
Lactate Erector spinae 140 min	1.38	−0.54
PCr/P_tot_	1.36	0.54
R^2^	0.32	
Q^2^	0.24	
CV-ANOVA *P*-value	0.020	
*n*	31	

VIP and p(corr) are reported for each regressor (i.e., the loading of each variable scaled as a correlation coefficient and therefore standardizing the range from −1 to + 1). The sign of p(corr) indicates the direction of the correlation with the dependent variable (+ = positive correlation; − = negative correlation). Hence, a positive p(corr) for a certain variable indicates a positive correlation with PPT of erector spinae in FM in the multivariate context. The four bottom rows report R^2^, Q^2^, *P*-value of the CV-ANOVA, and number of subjects included in the regression (*n*) from the OPLS regression (see Statistics for details).

**Table 9 jcm-09-03527-t009:** OPLS regressions of blood flow at baseline (140 min) in erector spinae in all subjects taken together, in CON, and in FM. Regressors used data from spectroscopy and MD data from baseline of this muscle (except pain variables from microdialysis). Only significant variables are shown (i.e., variables with VIP > 1.0 and absolute p(corr) ≥0.50).

All			CON			FM		
Variables	VIP	p(corr)	Variables	VIP	p(corr)	Variables	VIP	p(corr)
ATP	1.63	0.71	Pi/PCr	1.53	−0.68	Pi/PCr	1.87	−0.86
Pi/PCr	1.62	−0.71	ATP	1.42	0.64	Pi/P_tot_	1.69	−0.78
Pi/P_tot_	1.45	−0.64	Glycerol Erector spinae 140 min	1.39	−0.63	PCr/P_tot_	1.68	0.78
PCR	1.43	0.62	Pi/P_tot_	1.37	−0.61	PCR	1.17	0.54
PCr/P_tot_	1.42	0.62	PCr/P_tot_	1.20	0.54	ATP	1.14	0.52
R^2^	0.50		R^2^	0.56		R^2^	0.37	
Q^2^	0.45		Q^2^	0.47		Q^2^	0.25	
CV-ANOVA *p*-value	< 0.001		CV-ANOVA *P*-value	<0.001		CV-ANOVA *P*-value	0.022	
*n*	55		*n*	25		*n*	30	

VIP and p(corr) are reported for each regressor (i.e., the loading of each variable scaled as a correlation coefficient and therefore standardizing the range from −1 to + 1). The sign of p(corr) indicates the direction of the correlation with the dependent variable (+ = positive correlation; − = negative correlation). Hence, a positive p(corr) for a certain variable indicates a positive correlation with blood flow of erector spinae in the multivariate context. The four bottom rows of each regression report R^2^, Q^2^, *P* -value of the CV-ANOVA, and number of subjects included in the regression (*n*) from the OPLS regression (see Statistics for details).

## Data Availability

The datasets generated and/or analyzed in this study are not publicly available as the Ethical Review Board has not approved the public availability of these data.

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
