# Peer review of "Evidence of Mitochondrial Dysfunction in Fibromyalgia: Deviating Muscle Energy Metabolism Detected Using Microdialysis and Magnetic Resonance"

_jcm, 2020, doi:10.3390/jcm9113527_

Round 1

Reviewer 1 Report

It was a great pleasure to delve into the topic of fibromyalgia in a way designed by the authors of this publication. The work is perfectly divided, it is legible and clear in reception. Complicated, detailed and extensive diagnostics undertaken it this paper was logical and easy to receive. The range of samples tested is wide, which allows a more reliable approach to the topic of fibromyalgia. I am particularly impressed by the statistical approach - the conclusions thus drawn go beyond the standard results of similar studies. I fully agree with the authors of the publication who write themselves (in paragraph 4.7) that the strength of this work is in "combined traditional statistical methods with advanced multivariate methods to handle intercorrelated
variables". It makes this study not a hypothesis-driven, but a holistic one - given the biological aspect of pain in the disease which leads to the activated molecular mechanisms from a "broader system biology perspective" (as the authors mentioned themself in paragraph 4.7).

Thank You for the opportunity to get acquainted with this paper.

Author Response

Response to reviewer 1

It was a great pleasure to delve into the topic of fibromyalgia in a way designed by the authors of this publication. The work is perfectly divided, it is legible and clear in reception. Complicated, detailed and extensive diagnostics undertaken it this paper was logical and easy to receive. The range of samples tested is wide, which allows a more reliable approach to the topic of fibromyalgia. I am particularly impressed by the statistical approach - the conclusions thus drawn go beyond the standard results of similar studies. I fully agree with the authors of the publication who write themselves (in paragraph 4.7) that the strength of this work is in "combined traditional statistical methods with advanced multivariate methods to handle intercorrelated variables". It makes this study not a hypothesis-driven, but a holistic one - given the biological aspect of pain in the disease which leads to the activated molecular mechanisms from a "broader system biology perspective" (as the authors mentioned themself in paragraph 4.7).

Thank You for the opportunity to get acquainted with this paper.

Our answer: Thank you for the very positive and encouraging comments concerning our manuscript.

Reviewer 2 Report

The authors investigated the muscular mitochondrial dysfunction by methods of microdialysis and magnetic resonance spectroscopy in patients with fibromyalgia, and analyzed the association between the dysfunction and pain measures by multivariate statistical methods. As a result, alterations in mitochondrial functions-related metabolic substances were significantly correlated with pain measures. This study may support the already-known involvement of mitochondrial dysfunction in the pathophysiology of fibromyalgia. The reviewer would like following points to be considered.

1) Mitochondrial dysfunctions in patients with fibromyalgia were already suggested in previous studies, although the association between the dysfunction and pain symptoms have been inconsistent. In the introduction section, authors may want to clarify 1) What will this research provide new aspects regarding the pathophysiology of fibromyalgia, and 2) Why will this research put an end to the inconsistent (controversial) association.

2) This study employed the ACR criteria (1990). Authors seem to deny the new ACR criteria (2010, 2011) by describing as “These new criteria have had somewhat limited influence in the clinical milieu as many clinicians may consider hyperalgesia as important diagnostic information”.  Authors need to cite evidence to support this description.

3) Authors raised another reason to select the old criteria by describing “we wanted to compare our study with earlier studies, we chose the ACR criteria from 1990”. In this context, authors should clarify which criteria was used in studies cited for discussion in this study. Inclusion of studies with the new criteria in discussion would make a confusion.

4) How was the number of subjects (33 female patients) determined? Is this number sufficient to draw a conclusion?

5) The range of subject's age is relatively wide (22 – 56 years old) and the BMI in FM group is significantly higher than that in control group. Stratified analysis might be considered to control these possible confounding factors.

Author Response

Response to Reviewer 2

The authors investigated the muscular mitochondrial dysfunction by methods of microdialysis and magnetic resonance spectroscopy in patients with fibromyalgia,and analyzed the association between the dysfunction and pain measures by multivariate statistical methods. As a result, alterations in mitochondrial functions-related metabolic substances were significantly correlated with pain measures. This study may support the already-known involvement of mitochondrial dysfunction in the pathophysiology of fibromyalgia. The reviewer would like following points to be considered.

Our answer: The authors would like to thank Reviewer 2 for their time and their positive reception of our article. We are grateful for the helpful comments and suggestions. We have responded pointwise below to the reviewer’s comments (which are in bold).

  • Mitochondrial dysfunctions in patients with fibromyalgia were already suggested in previous studies, although the association between the dysfunction and pain symptoms have been inconsistent. In the introduction section, authors may want to clarify 1) What will this research provide new aspects regarding the pathophysiology of fibromyalgia, and 2) Why will this research put an end to the inconsistent (controversial) association.

Our answer: Point taken
In the end of the introduction we due to this comment have added the following:
Thus, this relatively large study using different methods to capture the muscle metabolic status has potentials to shed further light on pathophysiology in FM i.e. possible mitochondrial involvement. Moreover, this study applying advanced multivariate methods can with better precision than earlier studies determine if muscle metabolic alterations are associated with pain aspects in FM.”  

  • This study employed the ACR criteria (1990). Authors seem to deny the new ACR criteria (2010, 2011) by describing as “These new criteria have had somewhat limited influence in the clinical milieu as many clinicians may consider hyperalgesia as important diagnostic information”.  Authors need to cite evidence to support this description.

Our answer: Point taken.
This was just a speculation by us, and we have no evidence to support our statement. Due to this comment we have omitted the speculation from the sentence which now has the following wording:
“However, these new criteria have had somewhat limited influence in the clinical milieu.”

  • Authors raised another reason to select the old criteria by describing “we wanted to compare our study with earlier studies, we chose the ACR criteria from 1990”. In this context, authors should clarify which criteria was used in studies cited for discussion in this study. Inclusion of studies with the new criteria in discussion would make a confusion.

Our answer: Point taken.
In the revised version of the manuscript we in the discussion have added information which original studies that have used other criteria than the ACR 1990 criteria of FM. Most studies referred used the ACR 1990 criteria.
It must be noted that there is considerable overlap between the criteria as noted by several authors (Fors et al 2020; Galvez-Sanchez & Reyes del Paso 2020; even though the criteria have somewhat different emphasis. In epidemiological studies more recent criteria than those of 1990 appear to result in higher prevalence with a higher proportion of males (Sarzini-Puttini et al 2018).

  • How was the number of subjects (33 female patients) determined? Is this number sufficient to draw a conclusion?

Our answer: Point taken.
This is an important point. In the ethical application prior to start of this project we made power analysis both based upon the concentrations of lactate and spectroscopy data. Both these analyses indicated that 25 subjects in each group were necessary. Hence, the number of subjects is sufficient to draw conclusions. In fact, this is the largest study of patients with fibromyalgia participating in microdialysis and in spectroscopy. The earlier studies of fibromyalgia/widespread pain using these methods have used smaller number of subjects and (as reported in the manuscript) found significant differences in metabolites. Hence, we are convinced that the number of patients and controls are sufficient.

Due to this comment we have added the following sentence in section 2.1 Subjects: “The number of subjects were determined using Power and sample size calculation ver 3.0.2 (Dupont & Plummer) based upon results both from microdialysis (the concentration of lactate according to Rosendal et al) and from ATP concentration obtained from spectroscopy of vastus lateralis (Gerdle et al). Both analyses indicated that 25 subjects in each group were necessary.”  

  • The range of subject's age is relatively wide (22 – 56 years old) and the BMI in FM group is significantly higher than that in control group. Stratified analysis might be considered to control these possible confounding factors.

Our answer: Point mainly taken.
Due to this comment we have analyzed using OPLS if age and BMI together with group membership are significant regressors of concentrations of PCr, ATP, puruvate at baseline in both muscles. We found that besides group membership also BMI but not age was a significant regressor of PCr, ATP, puruvate of trapezius, respectively. No significant regression of pyruvate of erector spinae was obtained.    
We have added the following in section 4.5 of the discussion: ”Partly in agreement with these observations we found that besides group (FM or CON) BMI - but not age - was a significant regressor of PCr, ATP, and puruvate of trapezius according to OPLS regressions (data not shown).

Round 2

Reviewer 2 Report

In my opinion, the following sentences are not appropriate if author does not cite evidence. “However, these new criteria have had somewhat limited influence in the clinical milieu. Moreover, the newer criteria, including ACTTION-APS Pain Taxonomy (AAPT) (2018) [22], have not gained traction in the clinical setting.” It does not seem necessary to deny other criteria. It seems enough to describe just the reason why author used the ACR criteria (1990).

Author Response

In my opinion, the following sentences are not appropriate if author does not cite evidence. “However, these new criteria have had somewhat limited influence in the clinical milieu. Moreover, the newer criteria, including ACTTION-APS Pain Taxonomy (AAPT) (2018) [22], have not gained traction in the clinical setting.” It does not seem necessary to deny other criteria. It seems enough to describe just the reason why author used the ACR criteria (1990).

Our comments: We agree and have omitted these sentences.